# Tuning Multicolor Emission of Manganese-Activated Gallogermanate Nanophosphors by Regulating Mn Ions Occupying Sites for Multiple Anti-Counterfeiting Application

**DOI:** 10.3390/nano12122029

**Published:** 2022-06-13

**Authors:** Dangli Gao, Peng Wang, Feng Gao, William Nguyen, Wei Chen

**Affiliations:** 1College of Science, Xi’an University of Architecture and Technology, Xi’an 710055, China; wp2013141996@163.com (P.W.); gf@xauat.edu.cn (F.G.); 2Department of Physics, The University of Texas at Arlington, Arlington, TX 76019-0059, USA; william.nguyen@uta.edu

**Keywords:** Zn_3_Ga_2_GeO_8_:Mn phosphors, multicolor emission, hydrothermal approach, afterglow, anti-counterfeiting application

## Abstract

The ability to manipulate the luminescent color, intensity and long lifetime of nanophosphors is important for anti-counterfeiting applications. Unfortunately, persistent luminescence materials with multimode luminescent features have rarely been reported, even though they are expected to be highly desirable in sophisticated anti-counterfeiting. Here, the luminescence properties of Zn_3_Ga_2_GeO_8_:Mn phosphors were tuned by using different preparation approaches, including a hydrothermal method and solid-state reaction approach combining with non-equivalent ion doping strategy. As a result, Mn-activated Zn_3_Ga_2_GeO_8_ phosphors synthesized by a hydrothermal method demonstrate an enhanced red photoluminescence at 701 nm and a strong green luminescence with persistent luminescence and photostimulated luminescence at 540 nm. While Mn-activated Zn_3_Ga_2_GeO_8_ phosphors synthesized by solid-state reactions combined with a hetero-valent doping approach only exhibit an enhanced single-band red emission. Keeping the synthetic method unchanged, the substitution of hetero-valent dopant ion Li^+^ into different sites is valid for spectral fine-tuning. A spectral tuning mechanism is also proposed. Mn-activated Zn_3_Ga_2_GeO_8_ phosphors synthesized by a hydrothermal approach with multimodal luminescence is especially suitable for multiple anti-counterfeiting, multicolor display and other potential applications.

## 1. Introduction

The ability to tune the luminescent color of materials is essential for various applications, such as anti-counterfeiting, three-dimensional displays, information coding, bio-imaging, optoelectronic devices and luminescent labeling [1,2,3]. Conventional approaches involve utilizing specially designed organic dyes or quantum dots upon ultraviolet or blue light excitation, where the emission color is modulated by tuning the wavelengths or the power density of excitation light [4,5], thus posing limitations in the resolution of imaging due to auto-fluorescence. Persistent luminescence (PersL) phosphors, which can emit luminescence lasting for hours after the stoppage of the excitation light [6], are particularly suitable for such imaging applications as they emit no background fluorescence [7,8].

The transition metal Mn with the multiple oxidation states, e.g., +2, +3 and +4, provides an opportunity for multi-color emission [9]. In recent years, Mn-doped ZnGa_2_O_4_ microcrystals with unique luminescence features have attracted much attention because a ZnGa_2_O_4_ host has two kinds of stable chemical coordination structure, including Ga^3+^ sites with octahedral coordination and Zn^2+^ sites with tetrahedral coordination [10]. Generally, a Mn^2+^ activation center occupying the tetrahedral-coordinated sites shows a green emission with long PersL, while a Mn^4+^ activation center with an octahedral-coordinated structure demonstrates a luminescence emission from red to deep red. Apparently, the Mn activator shows a green to deep red emission, and the emission color is determined by the coordination environment of Mn ions in the crystal structure.

In our work, Mn-activated Zn_3_Ga_2_GeO_8_ phosphors are successfully prepared using a hydrothermal method and solid-state reaction approach. Interestingly, it was found that changing the preparation route was a more efficient method for the spectral tuning of Mn-activated Zn_3_Ga_2_GeO_8_ phosphors relative to a non-equivalent ion doping strategy. Mn-activated Zn_3_Ga_2_GeO_8_ phosphors show an enhanced red photoluminescence (PL) at 701 nm and a strong green emission at 540 nm with PersL and a green photostimulated luminescence (PSL) by Li^+^ substituted for Zn^2+^ or Ga^3+^ sites under hydrothermal conditions. Zn_3_Ga_2_GeO_8_ phosphors synthesized by a solid-state reaction only exhibit an enhanced pure-red broad-band luminescence. Particularly, Zn_3_Ga_2_GeO_8_ phosphors synthesized by a hydrothermal approach exhibit multicolor and multimode luminescence properties, which are especially suitable for multiple anti-counterfeiting and have a great potential for multicolor display, anti-forgery, and other potential applications.

## 2. Experimental Section

### 2.1. Chemicals

Ga_2_O_3_ (99.999%), GeO_2_ (99.999%), LiNO_3_ (99.9%), Mn(NO_3_)_2_·*x*H_2_O (99.9% metals basis), Zn(NO_3_)_2_·6H_2_O (99%), ZnO (99.99%), and MnO (99.99%) were obtained from Aladdin (Shanghai, China). Hydrochloric acid (AR, 36.0–38.0%), nitric acid (AR, 36.0–38.0%), sodium hydroxide, concentrated ammonium hydroxide, anhydrous ethanol and polyvinyl alcohol (PVA) with analytical grade are stocked from Sinopharm Chemical Reagent Co. Ltd. (Shanghai, China).

### 2.2. Sample Preparation

Zn_3_Li_0.4_Ga_1.6_GeO_8_:0.25% Mn phosphor was synthesized by using a hydrothermal method [8,11]. The synthesis procedure is described below. GeO_2_ powders were dissolved in a sodium hydroxide solution to achieve 0.5 M Na_2_GeO_3_ solution. Then, 3.0 mL 0.5 M Zn(NO_3_)_2_, 0.0375 mL 0.1 M Mn(NO_3_)_2_ and 0.3 mL concentrated HNO_3_ were together added slowly into 10 mL deionized water and then violently stirred. Subsequently, 0.4 mL 0.5 M LiNO_3_, 1.6 mL 0.5 M Ga(NO_3_)_3_ and 1.1 mL 0.5 M Na_2_GeO_3_ were added to the above solution. After the solution was stirred vigorously for 1 h, concentrated ammonium hydroxide was added into the mixture of precursor solution to tune the pH of the mixed solution to 7.5, and then the mixed solution was stirred vigorously for 1 h at ambient temperature. The final colloid was transferred into a 50 mL polytetrafluoroethylene reactor, which was then placed in an oven and heated at 220 °C for 10 h. Finally, the resulting suspension was centrifuged and washed 3 times using deionized water. The collected products were dried at 70 °C, and were then annealed in a chamber-type electric resistance furnace in air at 1100 °C for 2 h.

The synthesis procedure of Zn_2.4_Li_0.6_Ga_2_GeO_8_:Mn phosphors by hydrothermal method was similar to that of Zn_3_Li_0.4_Ga_1.6_GeO_8_:0.25% Mn only via 0.6 mL 0.5 M LiNO_3_ substituting for 20% Zn(NO_3_)_2_. According to our previous study [8,11], 20% Li substitution is an optimal doping concentration.

Zn_3_Ga_2_GeO_8_:0.25% Mn phosphors were prepared via a solid-state reaction approach using ZnO, Ga_2_O_3_, GeO_2_, and MnO as raw materials [12]. Raw materials were weighted on the basis of the formula of Zn_3_Ga_2_GeO_8_:0.25%Mn and finely mixed in an agate. All the ground powders were then pre-calcined at 900 °C for 2 h. Subsequently, the pre-calcined products were sintered at 1100 °C for 2 h and then cooled down to ambient temperature.

The synthesis procedure of Zn_2.4_Li_0.6_Ga_2_GeO_8_:Mn phosphor synthesized by a solid state reaction approach was similar to that of Zn_3_Ga_2_GeO_8_:0.25% Mn only via Li substituting for 20% ZnO.

### 2.3. Inks Preparation and Anti-Counterfeiting Patterns

Luminescence inks were fabricated based on a modified approach [13]. Typically, the prepared phosphors were dispersed into a mixed solution of 1.0 mL hydrochloric acid (0.2 mol/L) and 1.0 mL anhydrous C_2_H_5_OH, and then the mixed solution was centrifuged and washed with deionized water and anhydrous C_2_H_5_OH several times. Finally, the powders were mixed with PVA aqueous solution in a ratio of 1:1 to fabricate luminescent anti-counterfeiting inks. The luminescent patterns printed on the paper were achieved using screen printing method.

### 2.4. Characterization

A D/Max2400 X-ray diffractometer (XRD, Rigaku, Japan) with Cu Kα (40 kV, 100 mA) irradiation (λ = 1.5406 Å) was employed to characterize the crystal structures of phosphors. The shape and size of phosphors were characterized using a ZEISS Gemini 500 scanning electron microscopy (SEM) (Oberkochen, Germany). A spectrometer (PHI 5600ci ESCA, PerkinElmer, Waltham, MA, USA) with monochromatized Al K radiation was used to measure the X-ray photoelectron spectra (XPS) at room temperature. Thermoluminescence (TL) curves of products were measured using a TOSL-3DS TL spectrometer (Guangzhou, China) with a temperature range from 25 to 500 °C and a heating rate of 2 °C/s. A Horiba PTI QuantaMaster 8000 spectrofluorometer (Burlington, ON, Canada) equipped with a 75 W xenon lamp was used to study the optical properties of products. In addition, besides the xenon lamp as an irradiation source, two UV lamps (a 4-W 254 nm and a 5-W 365 nm) and two NIR laser diodes (0–2 W 808 nm and 0–5 W 980 nm) were also employed as excitation sources. When NIR lasers were used for irradiating the anti-counterfeiting patterns, the entire anti-counterfeiting pattern was programmatically scanned by the laser beam at a rate of 3 times per min. A Nikon EOS 60D camera (Tokyo, Japan) was used to take photographs of anti-counterfeiting patterns with suitable optical filters.

## 3. Results and Discussion

Figure 1 exhibits the XRD patterns of the four samples synthesized by different methods combined with a non-equivalent doping strategy. As shown in Figure 1b,e, the XRD patterns of the two samples (one is Zn_3_Li_0.4_Ga_1.6_GeO_8_:Mn synthesized by a hydrothermal approach, and the other is Zn_3_Ga_2_GeO_8_:Mn prepared by a solid-state reaction method) are similar and both dominated by a spinel-structured solid solutions of cubic phase ZnGa_2_O_4_ (JCPDS No. 01-071-0843) and cubic phase Zn_2_GeO_4_ (JCPDS No. 01-007-9080) (Figure 1f,g), accompanied with a rhombohedral Zn_2_GeO_4_ (JCPDS No. 04-007-5691) secondary phase (Figure 1a). Zn_2.4_Li_0.6_Ga_2_GeO_8_:Mn samples synthesized by a hydrothermal approach, as shown in Figure 1c, are the compounds of cubic ZnGa_2_O_4_ (JCPDS No. 01-071-0843) with an *Fd*$\bar{3}$*m* space group and rhombohedral Zn_2_GeO_4_ (JCPDS No. 04-007-5691) with an *R*$\bar{3}$ space group [6,14,15]. While the XRD patterns of Zn_2.4_Li_0.6_Ga_2_GeO_8_:Mn prepared by a solid-state reaction method in Figure 1d demonstrate spinel-structured solid solutions of ZnGa_2_O_4_ and Zn_2_GeO_4_ [16,17]. It is reported that Zn_3_Ga_2_GeO_8_ solid solutions are formed by Ge^4+^ substituting for Ga^3+^ into a ZnGa_2_O_4_ lattice, where excessive Zn^2+^ is necessary to counterbalance the charge disequilibrium [6,17]. It was observed that a solid solution is easier to form when Li replaces Ga, as shown in Figure 1b, which corresponds to a reduced Ga to Zn ratio but an excess of Zn in the hydrothermal reaction process, relative to Li^+^ replacing Zn^2+^ (Figure 1c). With the addition of Li^+^, substituting for Zn^2+^ (Figure 1d) in the solid-state reaction, the phase purity becomes higher relative to the results shown in Figure 1e, indicating that the Ge^4+^ ions (53 pm for ionic radius) enter into octahedral Ga^3+^ (62 pm for ionic radius) sites [17] and Li^+^ (76 pm for ionic radius) is substituted for Zn^2+^ (74 pm for ionic radius) sites [18] for charge compensation [19]. The results of XRD patterns indicate that different preparation approaches and non-equivalent doping strategies have a great influence on the crystal structure of the Zn_3_Ga_2_GeO_8_ phosphor.

SEM images (Figure 2) demonstrate that the morphology of these phosphors are irregular micro-particles with faceted surfaces. Their size remarkably changes from around 0.3 μm for Zn_3_Li_0.4_Ga_1.6_GeO_8_:Mn and Zn_2.4_Li_0.6_Ga_2_GeO_8_:Mn phosphors (Figure 2a,b) synthesized by a hydrothermal method to around 5 μm for Zn_2.4_Li_0.6_Ga_2_GeO_8_:Mn and Zn_3_Ga_2_GeO_8_:Mn (Figure 2c,d) synthesized by a solid-state reaction. Elemental mappings of Zn_3_Li_0.4_Ga_1.6_GeO_8_:Mn phosphors synthesized by a hydrothermal method and Zn_3_Ga_2_GeO_8_:Mn phosphors synthesized by a solid-state reaction (Figure 2e,f) show the even distributions of Zn, Ge, Ga, and O, indicating that there is no difference in element types and element distribution in these two kinds of samples. Clearly, the preparation method is one of the key factors that determines the crystal structure and morphology of these phosphors. The reasons for the differences in the crystal phase structure of the samples prepared by different methods could be complex. In a solid-state reaction system, a high temperature facilitates GeO_2_ volatilization, leading to the formation of a solid solution. On the other hand, hydrothermal conditions may also be beneficial for the coexistence of rhombohedral Zn_2_GeO_4_ and cubic ZnGa_2_O_4_.

All phosphors used in the experiments were annealed at 1100 °C due to their more distinctive luminescence features than their as-synthesized counterparts. Manganese-activated gallogermanate-based phosphors synthesized by a hydrothermal approach via Li replacing 20% Ga^3+^ are referred to as Zn_3_Li_0.4_Ga_1.6_GeO_8_:Mn phosphor, and those synthesized via Li replacing 20% Zn^2+^ are referred to as Zn_2.4_Li_0.6_Ga_2_GeO_8_:Mn phosphor. The Zn_3_Li_0.4_Ga_1.6_GeO_8_:Mn phosphor was investigated first, and Figure 3 illustrates its excited wavelength-dependent luminescence features. Under a 254 nm UV lamp excitation, a blue–white broad-band spectrum in the range of 400–750 nm was achieved (Figure 3a) and could be ascribed to the transitions of matrix defect levels. When the excitation wavelength was switched to 357 nm, the strong red emission of Mn^4+^ ions from ^2^E→^4^A_2_ (peaking at 701 nm) transitions could be obtained, accompanied with a weak green emission (540 nm) from ^4^T_1_(4G)→^6^A_1_(6S) transitions of Mn^2+^ ions (Figure 3a) [20,21]. Finally, when the sample was irradiated with blue visible light at 467 nm, only a pure-red single band at 650–800 nm from ^2^E→^4^A_2_ of Mn^4+^ was observed (Figure 3a) [21]. PL excitation (PLE) spectrum monitoring at 540 nm shows a typical Mn^2+^–O^2−^ charge transfer band (CTB) centered at 283 nm (Figure 3b) [22]. PLE spectrum monitoring at 701 nm shows two broad bands (excitation band I at 200−400 nm and excitation band II at 400–620 nm) peaking at 357 nm and 467 nm in Figure 3b. The excitation band I peaking at 357 nm originated from ^4^A_2_→^4^T_1_, while excitation band II peaking at 467 nm was assigned to the ^4^A_2_→^4^T_2_ transitions of Mn^4+^ ions [23]. The profile and peak position on the PLE spectra monitored at 701 and 540 nm are different, which indicates that red and green emissions come from different Mn emission centers. These spectra indicate that Li^+^ replacing Ga^3+^ provides an opportunity for Mn to occupy the octahedral Ga^3+^ sites. As a result, in Zn_3_Li_0.4_Ga_1.6_GeO_8_:Mn crystals, Mn occupies two kinds of positions: one is a tetrahedral Zn site, and the other is octahedral Ga site. These results are consistent with the above XRD analysis.

In addition, the Zn_3_Li_0.4_Ga_1.6_GeO_8_:Mn phosphor can demonstrate superior green PersL properties. After the removal of UV radiation, the green luminescence (monitored at 540 nm) of the Zn_3_Li_0.4_Ga_1.6_GeO_8_:Mn phosphor shows a long PersL signal (Figure 3c). A PersL spectrum achieved at 7 s of the decay is shown in the inset of Figure 3b. The similar PersL profile indicated that the PersL could be attributed to the Mn^2+^ ions. From Figure 3c, it is evident that Zn_3_Li_0.4_Ga_1.6_GeO_8_:Mn phosphor exhibited a good green PersL. Moreover, the excitation wavelength-dependent PersL duration is shown in Figure 3c, which may originate from the pre-irradiated wavelength-dependent trapping and detrapping [24]. Aside from PL and PersL, the UV pre-irradiated Zn_3_Li_0.4_Ga_1.6_GeO_8_:Mn phosphor also exhibited superior PSL capabilities, peaking at 540 nm and 670 nm (Figure 3a) under the illumination of a 980 nm laser diode, which is consistent with previous reports [25,26]. We still found that the PL and PSL emission spectra were slightly different (Figure 3a). We know that green luminescence (with PersL feature) and red luminescence (without PersL characteristics) originate from the Mn^2+^ and Mn^4+^ luminescent centers, respectively. Theoretically, red PSL should not be observed due to no red PersL. Surprisingly, we could still observe red luminescence, but no green luminescence was achieved when we emptied the traps of Zn_3_Li_0.4_Ga_1.6_GeO_8_:Mn phosphor and then the sample was excited under 980 nm, suggesting that the red fluorescence may be derived from the upconversion fluorescence of Mn ions [27]. These results show that red and green PL with different luminescent features should derive from the luminescent center with different coordination environments.

Interestingly, the fluorescent colors of Mn-activated Zn_3_Ga_2_GeO_8_ phosphor prepared by a hydrothermal approach could be further finely adjusted by substituting Zn^2+^ with Li^+^ to tune the occupancy rate of Mn ions in Zn^2+^ and Ga^3+^ sites. Figure 4 shows the luminescence characteristics of Zn_2.4_Li_0.6_Ga_2_GeO_8_:Mn phosphor prepared by a hydrothermal approach. As expected, the fluorescence emission color changed from green to yellow to red in response to the excitation wavelength (Figure 4a). Under excitation at 254 nm, the sample showed a broad-band emission spanning a range of 400–800 nm wavelength (Figure 4a), which is similar to the emission of Zn_3_Li_0.4_Ga_1.6_GeO_8_:Mn phosphor. When the excitation wavelength was switched to 337 nm, the green emission (Figure 4a), which can be assigned to the ^4^T_1_(4G)→^6^A_1_(6S) transition of Mn^2+^, dominated the emission, accompanying the weak red emission peaking at 701 nm [26]. PLE spectrum monitored at 540 nm exhibited twin peaks in a 200–400 nm wavelength range (Figure 4b), which were assigned to the Mn^2+^–O^2−^ CTB and the 5d→5d transition of Mn^2+^ with tetrahedral coordination [9,22]. Under the 413 nm excitation, the phosphors exhibited a narrowband red emission (Figure 4a) originating from the spin-forbidden ^4^T_1_(4G)→^6^A_1_(6S) transition of Mn^2+^ in the strong octahedral crystal field, which is different from the deep red emission of Mn^4+^ (Figure 3a) of Zn_3_Li_0.4_Ga_1.6_GeO_8_:Mn phosphor. The PLE spectrum monitored at 701 nm exhibited triplet peaks at 276, 413 and 575 nm (Figure 4b), which were assigned to the Mn^2+^–O^2−^ CTB (276 nm) and the 5d→5d Mn^2+^ transition (413 and 575 nm) [24]. Aside from wavelength-dependent PL properties, the UV pre-irradiated Zn_2.4_Li_0.6_Ga_2_GeO_8_:Mn phosphor also showed excellent green PersL (Figure 4c) and green PSL (Figure 4a) capability under the 980 nm laser diode illumination.

Zn_2.4_Li_0.6_Ga_2_GeO_8_:Mn and Zn_3_Ga_2_GeO_8_:Mn phosphors synthesized by solid-state reaction demonstrated completely different fluorescence characteristics. The PLE and PL spectra of these phosphors are illustrated in Figure 5. We found that Zn_3_Ga_2_GeO_8_:Mn phosphor exhibited a broad-band red emission at 650–800 nm from ^2^E→^4^A_2_ transition of Mn^4+^ ions, accompanied with the weak green color broad-band emission at 500–600 nm from matrix defect in Figure 5a. Meanwhile, Zn_2.4_Li_0.6_Ga_2_GeO_8_:Mn phosphor demonstrates a broad-band pure-red emission at 650–800 nm upon the 365 nm excitation, indicating that Li doping turns Mn^4+^ into the spin-allowed weak crystal field, leading to a broader red band emission. In addition, the perceptible color of luminescence from the two samples can be directly observed, as shown in the digital imaging signals (Figure 5b). Compared with the orange light of Zn_3_Ga_2_GeO_8_:Mn phosphors without Li, the Zn_2.4_Li_0.6_Ga_2_GeO_8_:Mn phosphor exhibits an enhanced pure-red emission from Mn^4+^ ions at 701 nm. The profile and peak position of PLE spectra (Figure 5a) are similar to the PLE spectra monitored at 701 nm as shown in Figure 3b, which indicates that the broad-band red luminescence for three samples (including Zn_2.4_Li_0.6_Ga_2_GeO_8_:Mn and Zn_3_Ga_2_GeO_8_:Mn phosphors synthesized by solid-state reaction and Zn_3_Li_0.4_Ga_1.6_GeO_8_:Mn phosphor synthesized by hydrothermal approach) come from the same Mn^4+^ emission centers.

Comparing the spectral characteristics of the four samples, we found that the emission peak at 701 nm and their PLE spectra show the same spectral profile and position in the three samples (Zn_3_Li_0.4_Ga_1.6_GeO_8_:Mn synthesized by a hydrothermal approach; Zn_2.4_Li_0.6_Ga_2_GeO_8_:Mn and Zn_3_Ga_2_GeO_8_:Mn phosphors synthesized by solid-state reaction). Comparing the XRD patterns of the three samples, we can find that all three samples have the same spinel-structured solid solutions. Combined with the spectral features, the broad red emission bands peaking at 701 nm are easily ascribed to ^2^E→^4^A_2_ transitions of Mn^4+^ [28,29], which occupied octahedral sites in a spinel-structured solid solutions of ZnGa_2_O_4_ and Zn_2_GeO_4_. While the emission spectrum peaking at 701 nm and its PLE spectrum of Zn_2.4_Li_0.6_Ga_2_GeO_8_:Mn synthesized by a hydrothermal approach (Figure 4a,b) are different from the other three samples. The spin-forbidden red narrow-band emission peaking at 701 nm can be ascribed to the ^4^T_1_(4G)→^6^A_1_(6S) transitions of Mn^2+^, which occupied the strong octahedral crystal field [30,31] in the cubic phase ZnGa_2_O_4_ lattice. Similarly, green luminescence with PersL and PSL features can be observed only in Zn_3_Li_0.4_Ga_1.6_GeO_8_:Mn and Zn_2.4_Li_0.6_Ga_2_GeO_8_:Mn phosphors with a rhombohedral Zn_2_GeO_4_ phase synthesized by a hydrothermal approach, indicating that the rhombohedral Zn_2_GeO_4_ is a good matrix for generating green afterglow of Mn^2+^-occupied tetrahedral sites. The broad emission band (400–750 nm) from Zn_3_Li_0.4_Ga_1.6_GeO_8_:Mn and Zn_2.4_Li_0.6_Ga_2_GeO_8_:Mn phosphor synthesized by a hydrothermal approach could be assigned to matrix defects under a 254 nm UV lamp excitation, which can be further verified by the PL and PLE spectra of undoped Zn_3_Li_0.4_Ga_1.6_GeO_8_ phosphor, as shown in Figure 6a.

As a result, the hydrothermal approach is an effective method for the preparation of afterglow phosphors. The stored energy of Zn_3_Li_0.4_Ga_1.6_GeO_8_:Mn and Zn_2.4_Li0_.6_Ga_2_GeO_8_:Mn PersL phosphors can also be triggered by heating, which helps to release the absorbed energy and provide insights into PersL mechanism. TL curves monitored at 540 nm are shown in Figure 6b. The TL curves for Zn_3_Li_0.4_Ga_1.6_GeO_8_:Mn (Figure 6b, green curve) and Zn_2.4_Li_0.6_Ga_2_GeO_8_:Mn (Figure 6b, red curve) are similar and can be divided into two broad bands at 358 K and 376 K, and at 363 K and 423 K, respectively. These results indicate the presence of a shallow trap and deep trap in the two matrix materials, and the existence of the deep traps provides the conditions for generating PersL. Compared to Zn_3_Li_0.4_Ga_1.6_GeO_8_:Mn phosphor, the traps of Zn_2.4_Li_0.6_Ga_2_GeO_8_:Mn phosphor are deeper, which may explain why Zn_2.4_Li_0.6_Ga_2_GeO_8_:Mn phosphor has a long green afterglow duration. The possible luminescence mechanisms, including green PL, PersL and PSL from Mn^2+^, and the two red PL from Mn^2+^ and Mn^4+^ are proposed and shown in Figure 7 [8,22,32,33].

As shown in Figure 7, under UV excitation, electrons are first excited to the Mn^2+^–O^2−^ CTB or the excited state of Mn ions, some electrons reach the conduction band through photoelectric separation and are trapped by traps, and some electrons are relaxed to ^4^T_1_ and ^2^E levels through lattice vibration or non-radiation, resulting in ^4^T_1_ (4G)→^6^A_1_ (6S) (540 nm) transitions of the Mn^2+^-occupied tetrahedral site, ^4^T_1_ (4G)→^6^A_1_ (6S) (701 nm) transitions of Mn^2+^ at the octahedral site, and ^2^E→^4^A_2_ (701 nm) transitions of Mn^4+^ at the octahedral site. When the excitation is stopped, the trapped electrons in the trap reach the conduction band through lattice thermal vibration and are released to the ^4^T_1_ energy level of Mn^2+^ ions, resulting in green continuous fluorescence from ^4^T_1_ (4G)→^6^A_1_ (6S) (540 nm).

The luminescent emission color response to excitation wavelength and excitation power is convenient for various applications [8,34,35]. To verify the feasibility of the phosphors in the anti-counterfeiting fields, we used these phosphors as inks printed the table lamp patterns in Figure 8. Figure 8a depicts the schematic diagram of these patterns. The lampshade and bulb are printed with Zn_3_Li_0.4_Ga_1.6_GeO_8_:Mn and Zn_2.4_Li_0.6_Ga_2_GeO_8_:Mn phosphor synthesized by a hydrothermal approach, respectively. 

Upon the irradiation of 254 nm UV light, the lampshade became blue–white due to the color mixture of the Mn PL and the defect fluorescence of matrix [36] (Figure 8b), while the bulb became blue–green (Figure 8b). After the stoppage of excitation light, both the lampshade (dark green) and the bulb (bright green) emitted green PersL (Figure 8c), while upon the irradiation of 365 nm UV light, the lampshade and the bulb emitted bright red PL and green PL, respectively (Figure 8d). After the stoppage of UV excitation, these two patterns both showed green PersL colors with different saturations (Figure 8e). After these two patterns disappeared, the entire ‘desk lamp’ pattern was illuminated by a 980 nm laser diode; the lampshade and the bulb were lit again (Figure 8f), with the lampshade emitting a dark green color and the bulb emitting a yellow–green color (for a 980 nm illumination at 0.5 W), which is consistent with spectral characterization. Therefore, Mn-activated Zn_3_Ga_2_GeO_8_ phosphors hold a promise for multi-chrome (green, yellow and red) and multi-mode (PL, PersL, and PSL) anti-counterfeiting applications [37,38,39]. In addition, the PSL and PL of nanomaterials can be used for photodynamic activation [40], dosimetry [41], thermometry [42] and solid-state lighting [43].

Additionally, we need to point out that, for practical applications, the possibility of the power of the laser diodes (2 and 5 W for the 808 nm and 980 nm lasers, respectively) to generate even secondary heating effects on the samples should be considered. It is true that NIR lasers, such as 808 nm or 980 nm lasers, have been widely used for photothermal therapy (PTT) [44,45,46]. However, for PTT, the materials have strong absorptions for NIR, but they do not have luminescence. So, the energy absorbed is released as heat, while in the luminescence materials, the absorbed energy is released as luminescence [47,48]; therefore, heating is not a critical issue, even though it is unavoidable. Heating is always an issue for many applications with NIR lasers.

## 4. Conclusions

In summary, manganese-activated gallium germanate phosphors were successfully synthesized in two approaches, including a high-temperature solid-state reaction approach and a hydrothermal method. We found, for the first time, that these Mn-activated Zn_3_Ga_2_GeO_8_ phosphors prepared by a hydrothermal method and solid-state reaction method have different crystal phase structures and spectral properties. The phosphors prepared by the hydrothermal method exhibited a double-peak emission, including green PL with a PersL feature and red PL without a PersL effect, due to simultaneously having two kinds of occupancy sites. While the Mn-activated Zn_3_Ga_2_GeO_8_ phosphors prepared by a solid-state reaction method only exhibited red PL. In addition, Li^+^ ions were selectively substituted for the tetrahedral Zn site or the octahedral Ga site, leading to green light emission at 540 nm (^4^T_1_(G)→^6^A_1_(S) transition of Mn^2+^) with an afterglow and red light centered at 701 nm (^2^E→^4^A_2_ transition of Mn^4+^) in the absence of afterglow. Particularly, Mn-activated Zn_3_Ga_2_GeO_8_ phosphors synthesized by a hydrothermal approach via Li^+^ replacing the Zn^2+^ or Ga^3+^ sites exhibit dynamic and multicolor emissions as luminescence labels for multiple anti-counterfeiting, thus revealing the great potential of these phosphors in multicolor display, anti-forgery, and other potential applications.

## Figures and Tables

**Figure 1 nanomaterials-12-02029-f001:**
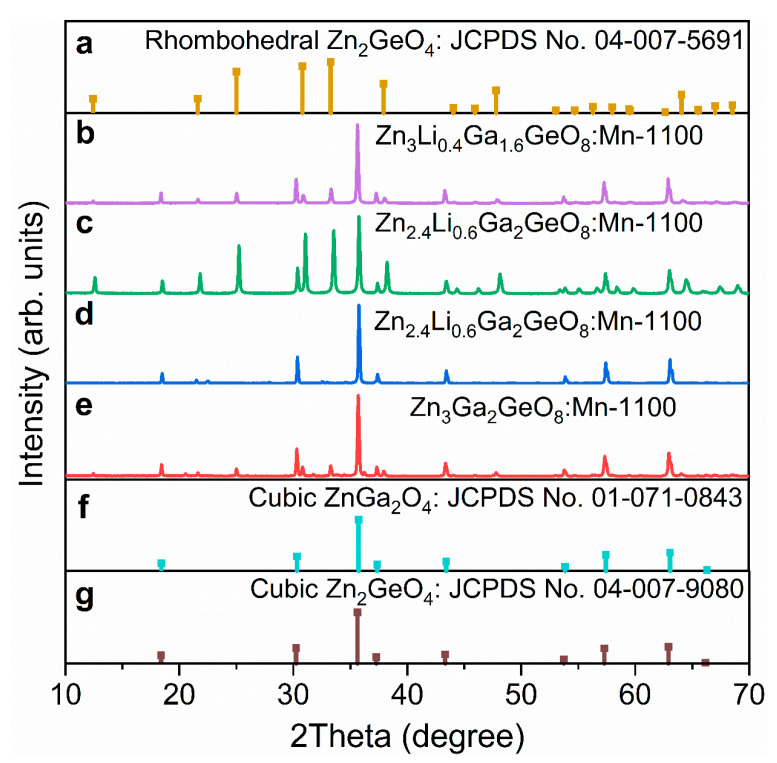
XRD patterns of Zn_3_Li_0.4_Ga_1.6_GeO_8_:Mn, Zn_2.4_Li_0.6_Ga_2_GeO_8_:Mn and Zn_3_Ga_2_GeO_8_:Mn phosphors. (**a**) Standard data for rhombohedral Zn_2_GeO_4_ (JCPDS No. 04-007-5691); (**b**,**c**) Zn_3_Li_0.4_Ga_1.6_GeO_8_:Mn and Zn_2.4_Li_0.6_Ga_2_GeO_8_:Mn phosphors synthesized by hydrothermal approach; (**d**,**e**) Zn_2.4_Li_0.6_Ga_2_GeO_8_:Mn and Zn_3_Ga_2_GeO_8_:Mn phosphors synthesized by solid-state reaction; (**f**,**g**) Standard data for cubic ZnGa_2_O_4_ (JCPDS No. 01-071-0843) and cubic Zn_2_GeO_4_ (JCPDS No. 04-007-9080).

**Figure 2 nanomaterials-12-02029-f002:**
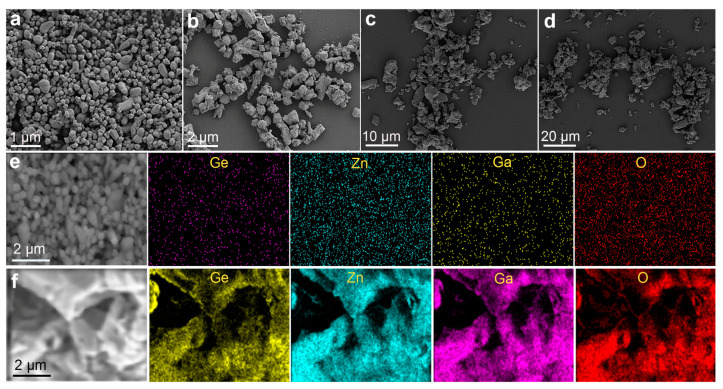
SEM images (**a**–**d**) and EDX mapping (**e**,**f**) of Zn_3_Ga_2_GeO_8_ phosphors annealed at 1100 °C. (**a**,**b**) Zn_3_Li_0.4_Ga_1.6_GeO_8_:Mn phosphor in (**a**) and Zn_2.4_Li_0.6_Ga_2_GeO_8_:Mn in (**b**) synthesized by hydrothermal approach; (**c**,**d**) Zn_2.4_Li_0.6_Ga_2_GeO_8_:Mn in (**c**) and Zn_3_Ga_2_GeO_8_:Mn in (**d**) synthesized by solid-state reaction; (**e**) EDX mapping of Zn_3_Li_0.4_Ga_1.6_GeO_8_:Mn synthesized by hydrothermal approach; and (**f**) EDX mapping of Zn_3_Ga_2_GeO_8_:Mn synthesized by solid-state reaction.

**Figure 3 nanomaterials-12-02029-f003:**
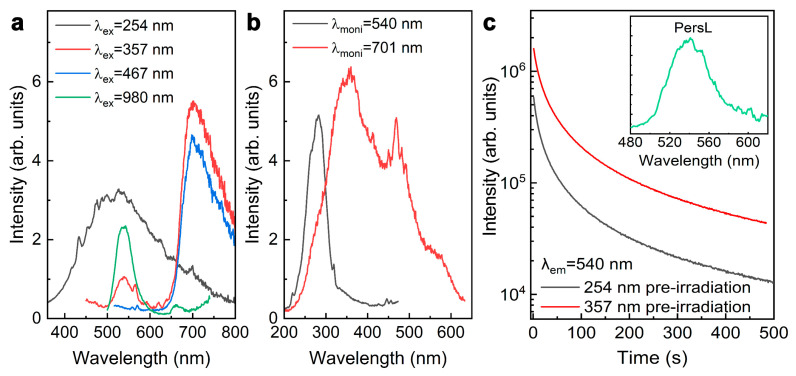
Excited wavelength-dependent fluorescence characteristics of Zn_3_Li_0.4_Ga_1.6_GeO_8_:Mn phosphors synthesized by hydrothermal approach. (**a**) PL spectra under various selective excitation and PSL upon 980 nm laser diode (0.5 W) irradiation; (**b**) PLE spectra (monitored at 540 or 701 nm); (**c**) PersL decay curves (monitoring at 540 nm) after irradiation by using 254 nm or 357 nm light for 5 min, the inset is the PersL emission spectrum achieved at 7 s of the decay.

**Figure 4 nanomaterials-12-02029-f004:**
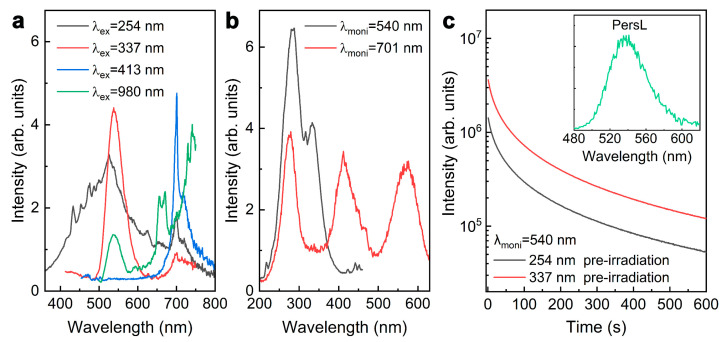
Excited wavelength-dependent fluorescence characteristics of Zn_2.4_Li_0.6_Ga_2_GeO_8_:Mn phosphors prepared by hydrothermal approach. (**a**) PL spectra under various selective excitation and PSL upon 980 nm laser diode (0.5 W) irradiation; (**b**) Excitation spectra monitoring at 540 and 701 nm; (**c**) PersL decay curves (monitoring at 540 nm) after irradiation by using 254 nm or 337 nm light for 5 min, the inset is the PersL emission spectrum achieved at 7 s of the decay.

**Figure 5 nanomaterials-12-02029-f005:**
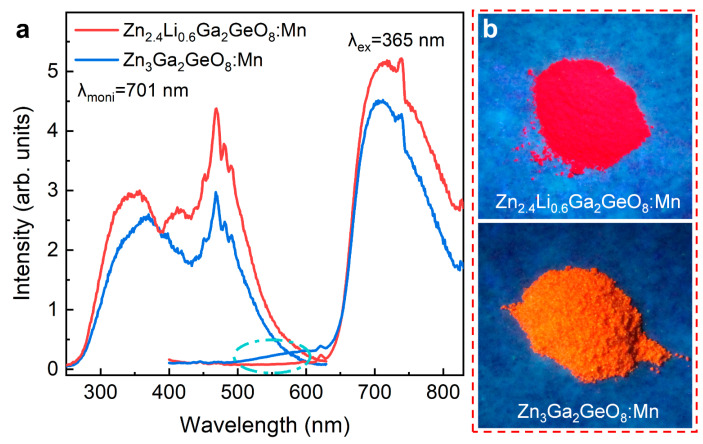
PLE and PL spectra and the corresponding digital PL images of Zn_2.4_Li_0.6_Ga_2_GeO_8_:Mn and Zn_3_Ga_2_GeO_8_:Mn phosphors synthesized by solid-state reaction. (**a**) PLE spectra monitored at 701 nm and PL spectra under 365 nm UV lamp excitation; (**b**) Digital PL photographs under 365 nm irradiation. The camera parameters are manual/ISO 3200/1 s. The slight difference in PL spectra can be seen in the cyan dash dot ellipse.

**Figure 6 nanomaterials-12-02029-f006:**
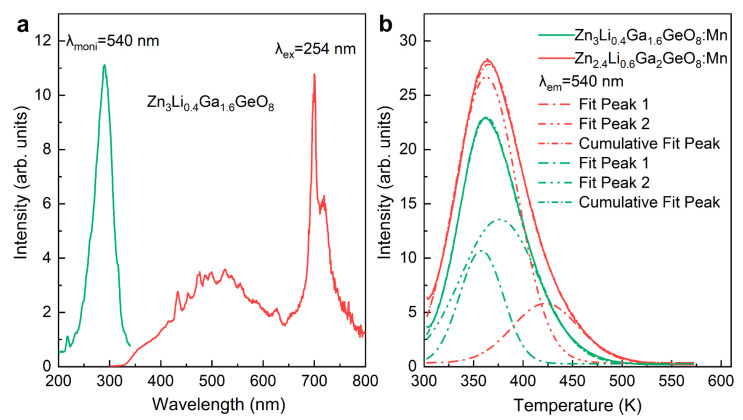
(**a**) PL (excitation at 254 nm) and PLE (monitoring at 540 nm) spectra of undoped Zn_3_Li_0.4_Ga_1.6_GeO_8_ phosphors synthesized by hydrothermal approach; (**b**) TL spectra obtained after 5 min illumination for Zn_3_Li_0.4_Ga_1.6_GeO_8_:Mn (green curve) and Zn_2.4_Li_0.6_Ga_2_GeO_8_:Mn phosphors (red curve) prepared by hydrothermal method.

**Figure 7 nanomaterials-12-02029-f007:**
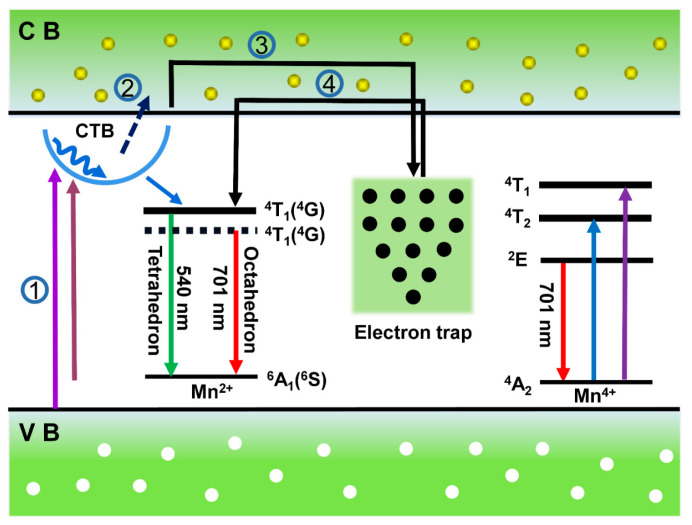
The proposed PL, PersL and PSL schematic diagram for green and red luminescence of Mn^2+^/Mn^4+^ in the Zn_3_Ga_2_GeO_8_ phosphors. Therein, ① UV light excitation, ② energy (or electron) transfer processes, ③ trapping, ④ release. The straight-line arrows and curved-line arrows stand for optical transitions and energy (or electron) transfer processes, respectively. Note that the solid and dashed lines represent the ^4^T_1_ energy levels of Mn^2+^ in the weak tetrahedral crystal field and strong octahedral crystal field, respectively.

**Figure 8 nanomaterials-12-02029-f008:**
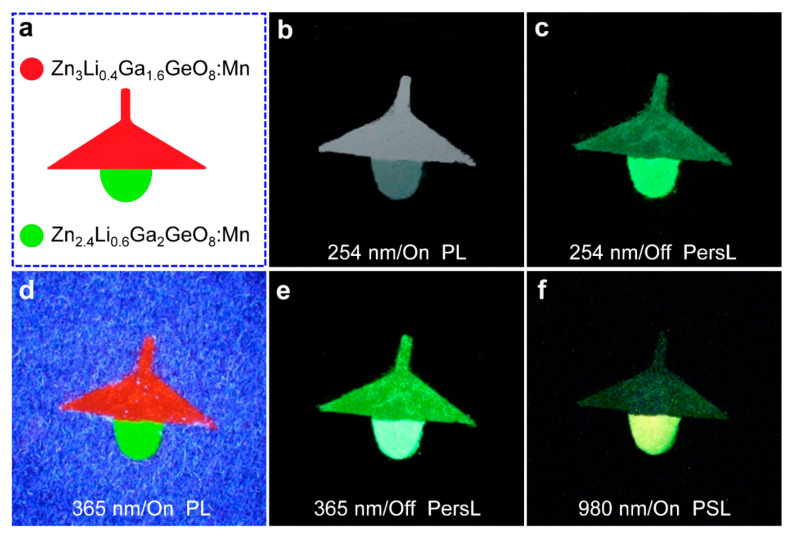
Digital photographs of a triple-mode (PL, PersL, and PSL modes) ‘chandelier’ pattern printed used Zn_3_Li_0.4_Ga_1.6_GeO_8_:Mn phosphor (lampshade) and Zn_2.4_Li_0.6_Ga_2_GeO_8_:Mn phosphor (light bulb) synthesized by hydrothermal approach. (**a**) The design of the ‘chandelier’ pattern; (**b**,**c**) PL and PersL images acquired upon and after 254 nm UV light excitation (for 5 min); (**d**,**e**) PL and PersL images acquired upon and after 365 nm UV light excitation (for 5 min); (**f**) PSL images upon a 980 nm laser diode irradiation (at 0.5 W). The PSL imaging was achieved until the disappearance of chandelier to the naked eye. The camera parameters are manual/ISO 3200/4 s.

## Data Availability

Not applicable.

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
