# Peer review of "Tuning Multicolor Emission of Manganese-Activated Gallogermanate Nanophosphors by Regulating Mn Ions Occupying Sites for Multiple Anti-Counterfeiting Application"

_nanomaterials, 2022, doi:10.3390/nano12122029_

Round 1

Reviewer 1 Report

See attached Comments (PDF file)

Author Response

Response to Reviewer 1 Comments

Point 1: Solid solution

OK! The reviewer understood that Zn1+xGa2−2xGexO4 is a complex material system. By the way, is it possible to show the JCPDS card plots of both “cubic-” and “hexagonal type” Zn3GaGexO4 in Fig. 1? [It seems that Figs. 1a and 1b correspond to those for the hexagonal Zn3GaGexO4 crystals (Zn2GeO4 type), whereas Figs. 1c and 1d correspond to those for the cubic ones (ZnGa2O4 type).]

Response 1: Thanks and Yes, the standard data of rhombohedral (hexagonal) and cubic Zn2GeO4 are shown in Figs. 1a and 1g, respectively.

Point 2: Figs. 1, 3, 4, 5, and 6, Please insert “space” between “a.” and “b.” like (a. u.) on the vertical-axis title units [because a. u. should be an abbreviation of arb. units (or is it “atomic units”?)].

Response 2: Thanks for the comments. The corresponding revisions have been made in the revised manuscript (See Figs.1, 3, 4, 5 and 6).

Point 3: Fig. 6 Please see below. There is no clear difference in the PL spectra between the two samples in Fig. 6a; however, their digital PL images in Fig. 6b are clearly different (upper image=pure red; lower image=orange like). Is it OK?

Response 3: Thanks for the comment. It is OK. The slight differences on PL spectra (in range of 500-650 nm) of Zn3Ga2GeO4:Mn and Zn2.4Li0.6Ga2GeO4:Mn phosphors were shown in the cyan dash dot ellipse. Zn3Ga2GeO4:Mn phosphor exhibits a weak broaden green emission at 500-650 nm and a strong red emission band at 650-800 nm, leading to the digital PL images with orange colors. While Zn2.4Li0.6Ga2GeO4:Mn phosphor only exhibits a strong red emission band at 650-800 nm. Our eyes and the camera have different sensitivies at different colors.

Point 4: Language

(1) We can find a word “respectively” in some places. Please note that a sentence “samples A and B were synthesized by the hydrothermal method and solid state reaction, respectively” may exactly mean “sample A was synthesized by the hydrothermal method, and sample B was synthesized by the solid state reaction.” In this sense, the reviewer cannot understand several sentences containing “respectively.” See, e.g., the sentence in p. 1, l. 14 “Here, Mn-activated Zn3Ga2GeO8 phosphors ... .” Was Mn-activated Zn3Ga2GeO8 phosphor without tunable color output synthesized by the hydrothermal method, and the same phosphor, but with tunable color output, synthesized by the solid state reaction?

See also p. 2, l. 49. Perhaps, no need of “respectively” in its sentence. Please also check a sentence containing “respectively” in p. 2, l. 81.

Response 4: Thanks for the comment. Corresponding modifications have been made in the revised manuscript.

Point 5: Language (2) English seems to be generally good, but some places contain errors. Please check or amend considering some examples below.

  1. 1
  2. 9 What is color output? The reviewer cannot understand “what is output of color.” Is it “light wavelength”?
  3. 10 luminescence (PersL) materials ® luminescence materials (because of no use of its abbreviation in Abstract)
  4. 14 and solid state reaction, respectively, via ® and solid state reaction, via (?)
  5. 15 the Ga site or the Zn site ® the Ga or Zn site
  6. 17 phosphors (synthesized ® phosphors synthesized
  7. 18 or Ga3+ sites shown ® or Ga3+ site showed
  8. 19 phosphors (synthesized by ® phosphors synthesized by
  9. 21 substituted Zn2+ or Ga3+ sites) exhibit ® substituted for Zn2+ or Ga3+ site exhibited
  10. 34 density of the excitation light ® intensity (?) of the excitation light (Note that “density of the light” is physically strange.)
  11. 2
  12. 45 from an orange to red. ® from a red to a deep red. (?)
  13. 51 substituted Zn2+ or ® substituted for Zn2+ or
  14. 54 substituted Zn2+ or ® substituted for Zn2+ or
  15. 2 (2. Experimental)

This section has those of “present” and “past” tense sentences. documents. Therefore, please unify basically with past tense sentences only. For example, “GeO2 powders are dissolved in ... .” in l. 67 should be “GeO2 powders were dissolved in ... .” l. 87 & 88 2 hours ® 2 h

Response 5: Thanks a lot for excellent suggestions. The corresponding revisions have been made in the revised manuscript.

  1. 9 The ”color output” were revised to “emission wavelength tuning”.
  2. 34 “the density of the excitation light” were revised to “the power density of the excitation light”.

Reviewer 2 Report

The authors have amended the manuscript according to the suggestions.

Author Response

Response to Reviewer 2 Comments

Point 1: The authors have amended the manuscript according to the suggestions.

Response 1: Thanks for your positive comment.

Reviewer 3 Report

The paper "nanomaterials-1741512" deals with the optical and structural properties of Mn-doped Gallogermanate nano phosphors. 

The experimental findings are relatively interesting but they seem very similar to previously published results by the same authors (ref 8 and, mainly 11).

On the other hand, this paper is not very clear, the authors report two synthesis methods, and two different Li doping percentages to compare with the un-doped samples. Further, the optical emission is studied as a function of at least 3 excitation wavelengths. With so many structural parameters and experimental setups is very difficult to create an ordered discussion and, unfortunately, the authors fail to clearly indicate their results.

As an example, figure 4a is mentioned in the discussion between figure 3a and figure 3b, while figure 4b seems to be not related to figure 4a (different samples and different optical measurements -PL,PLE vs thermoluminescence).

The sentence in line 126 "In addition, with the adding of Li+ for substituting Zn2+ (Figure 1c), the phase purity becomes higher and higher relative to Figure 1d, indicating the Ge4+ ions enter into octahedral Ga3+ sites by the Li+-Ge4+ co-substitution of Zn2+ and Ga3+ ions pairs for charge compensation" is not clear and the cited reference does not provide a very clear proof of the mentioned results (Lithium is not utilized in that paper).

Figure 6 refers to results reported in figure 3a, citing the similar results between "three samples", but there are 2 samples with the same broadband in figure 3a and the other two in figure 6...it is not obvious what the authors want to compare. Secondly,  figure6a reports very similar PLE and PL between the two samples while figure 6b reports a digital PL photography (under excitation at 365nm) of the same sample with a remarkable difference in the perceived color. This part is not clear.

Finally, the discussion about the role of Zn to define the Mn occupancy sites (lines 281-289) is not clear and should be rewritten and not only referenced to previous papers. 

Due to the above criticisms, I cannot recommend the paper to be published in the present form. The experimental results have the potential to be interesting but the paper needs a very deep revision to be clearly understandable by expert readers.

Author Response

Response to Reviewer 3 Comments

Point 1: The experimental findings are relatively interesting but they seem very similar to previously published results by the same authors (ref 8 and, mainly 11).

On the other hand, this paper is not very clear, the authors report two synthesis methods, and two different Li doping percentages to compare with the un-doped samples. Further, the optical emission is studied as a function of at least 3 excitation wavelengths. With so many structural parameters and experimental setups is very difficult to create an ordered discussion and, unfortunately, the authors fail to clearly indicate their results.

Response 1: Thanks for your comments. In this manuscript, luminescence properties of Zn3Ga2GeO8:Mn phosphors were tuned by using different preparation approach (including hydrothermal method and solid-state reaction approach) accompanied with hetero-valence ion doping strategy. As a result, the sample synthesized by hydrothermal way demonstrated multi-color and multi-mode fluorescence properties. While phosphors synthesized by solid-state reaction only exhibit an enhanced pure red single-band emission. The corresponding luminescence tuning mechanism is proposed.

Ref. 8 reported the achievement of quintuple-mode, multi-colour dynamic anti-counterfeiting using multi-mode Zn3Ga2GeO8:Cr3+,Yb3+,Er3+ and Zn1.6Li0.4GeO4:Mn2+ persistent phosphors as the luminescent inks. The design strategy reported in Ref. 8 may lead to the design and development of more sophisticated optical anti-counterfeiting technology.

Ref. 11 successfully synthesized Zn2GeO4: Mn,x%Li (x=0, 20, 50, and 70) and NaLiGe4O9:Mn micro-phosphors with the multi-chromatic and multi-mode luminescence via a hydrothermal approach followed by an annealing treatment. This work proves the feasibility of Li+ doping strategy in emission tuning, which can stimulate further studies on multi-mode luminescent materials in anti-counterfeiting applications.

All in all, they are three different works on three different material systems.

Point 2: As an example, figure 4a is mentioned in the discussion between figure 3a and figure 3b, while figure 4b seems to be not related to figure 4a (different samples and different optical measurements -PL,PLE vs thermoluminescence).

Response 2: Thanks. The logical order of the articles has been fully adjusted in the revised manuscript.

Point 3: The sentence in line 126 "In addition, with the adding of Li+ for substituting Zn2+ (Figure 1c), the phase purity becomes higher and higher relative to Figure 1d, indicating the Ge4+ ions enter into octahedral Ga3+ sites by the Li+-Ge4+ co-substitution of Zn2+ and Ga3+ ions pairs for charge compensation" is not clear and the cited reference does not provide a very clear proof of the mentioned results (Lithium is not utilized in that paper).

Response 3: Thank you!. The modifications have been made in the revised manuscript.

Point 4: Figure 6 refers to results reported in figure 3a, citing the similar results between "three samples", but there are 2 samples with the same broadband in figure 3a and the other two in figure 6...it is not obvious what the authors want to compare. Secondly,  figure6a reports very similar PLE and PL between the two samples while figure 6b reports a digital PL photography (under excitation at 365nm) of the same sample with a remarkable difference in the perceived color. This part is not clear.

Response 4: Thanks for your comment. Comparing Fig. 3, 4 and 5, we can conclude that the broadband spectrum originates from Mn4+ ions occupying octahedral lattice sites of Zn3Ga2GeO4 solid solution phase and the green emission originates from Mn2+ occupied tetrahedral lattice sites of rhombohedral phase Zn2GeO4.

In figure 6, the slight differences on PL spectra (in range of 500-650 nm) of Zn3Ga2GeO4:Mn and Zn2.4Li0.6Ga2GeO4:Mn phosphors were shown in the cyan dash dot ellipse. Zn3Ga2GeO4:Mn phosphor exhibits a weak broaden green emission at 500-650 nm and a strong red emission band at 650-800 nm, leading to the digital PL images with orange colors. While Zn2.4Li0.6Ga2GeO4:Mn phosphor only exhibits a strong red emission band at 650-800 nm. Note that the stimulation effect on the camera and human eye of the red fluorescence intensity of ten times is simillar to the green fluorescence intensity [S. J. Robinson, J. T. Schmidt, Mater. Eval. 1984, 42, 1029].

Point 5: Finally, the discussion about the role of Zn to define the Mn occupancy sites (lines 281-289) is not clear and should be rewritten and not only referenced to previous papers.

Response 5: A good question. The discussion about the role of Zn to define the Mn occupancy sites was rewritten (XRD analysis at lines 137-147, spectral analysis at lines 304-329) in the revised manuscript.

Round 2

Reviewer 3 Report

The paper was improved and now it sounds much more clear. 

In my opinion the presentation of the results in figure 3 and figure 4 can be improved for a more clear presentation of the data, but my comments can be considered as minor indication.

It would much more interesting if the author give some more details on the laser power density utilized for NIR excitation. They just indicate the power of the laser diodes (2 and 5 W for the 808 and 980, respectively ) that could generate even secondary heating effects on the samples and/or up conversion effects.

Author Response

Response to Reviewer 3 Comments

Point 1: The paper was improved and now it sounds much more clear. 

In my opinion the presentation of the results in figure 3 and figure 4 can be improved for a more clear presentation of the data, but my comments can be considered as minor indication.

It would much more interesting if the author give some more details on the laser power density utilized for NIR excitation. They just indicate the power of the laser diodes (2 and 5 W for the 808 and 980, respectively) that could generate even secondary heating effects on the samples and/or up conversion effects.

Response 1: We thank the Reviewer for the comment. We have revised Fig. 3 and 4 in the revised manuscript for a more clear presentation of the data. We gave some more details on the laser power density utilized for NIR excitation in Fig. 3 and 4.

Regarding the possibility of the power of the laser diodes (2 and 5 W for the 808 and 980, respectively) may generate even secondary heating effects on the samples and/or up conversion effects is a very interesting question. Yes, it is true that the NIR lasers like 808 nm or 980 nm have been widely used for photothermal therapy (PTT), However, for PTT, the materials have strong absorptions in NIR but they do not have luminescence. So, the energy absorbed would be released as heat, while in our luminescence materials, the absorbed energy would be released as luminescence, therefore, heating is not a critical issue even though it is not avoidable. Heating is always an issue for many applications with NIR lasers.

We added these comments in the revision   

This manuscript is a resubmission of an earlier submission. The following is a list of the peer review reports and author responses from that submission.

Round 1

Reviewer 1 Report

In this work, the authors prepared Mn-doped Zn3Ga2GeO8 via hydrothermal and solid-state methods and investigated its luminescence properties. They substituted Li+ for the Zn2+ or Ga3+ sites and showed the dependency of PL and PSL on the excitation wavelength. Also, the resulting phosphor The results reported are interesting and well organized with proper discussion. But, there are several minor issues to be addressed before going to the publication.

  1. In this work, Li ions were replaced for Zn or Ga and the PL data were shown in Fig. 3 and Fig. 4. In terms of achieving improved photoluminescence, which case is better? That is, there is no direct comparison in luminescent properties between Zn2.8Li0.2Ga2GeO8:Mn and Zn3Li0.2Ga1.8GeO8:Mn. The comparison is helpful for readers who have interest. Thus, if possible, compare the PL properties between Zn2.8Li0.2Ga2GeO8:Mn and Zn3Li0.2Ga1.8GeO8:Mn, and give some discussion.

  1. In addition, there is no mention about the excellency of the samples prepared by the hydrothermal method compared with that prepared by the solid-state reaction. Give a short comment about this with comparing the PL intensity.

  1. In Fig. 5, the PLE spectra are not matched with the PL spectra. At 365 nm, the PLE intensity for the sample without Li+ is higher than that with Li+. But, in the PL emission, the Li-substituted sample is better than the sample without Li+. Therefore, it is necessary to check if the data is correct.

  1. Is there any reason that both Li and Mn contents are fixed?

  1. In XRD patterns, the Li content is given as 20%. Actually, the Li+ concentration used is about 6.7% of Zn and about 10% of Ga. Check again the Li+ concentration.

  1. There is no information about the doping place of Mn. Authors should give the exact information about the site where Mn ions are substituted.

  1. Line 111 and 115, the ‘PDF#’ expression is needed to be changed as JCPDS #.

Author Response

Reviewer #1:

In this work, the authors prepared Mn-doped Zn3Ga2GeO8 via hydrothermal and solid-state methods and investigated its luminescence properties. They substituted Li+ for the Zn2+ or Ga3+ sites and showed the dependency of PL and PSL on the excitation wavelength. Also, the resulting phosphor. The results reported are interesting and well organized with proper discussion. But, there are several minor issues to be addressed before going to the publication.

 Comment 1: In this work, Li ions were replaced for Zn or Ga and the PL data were shown in Fig. 3 and Fig. 4. In terms of achieving improved photoluminescence, which case is better? That is, there is no direct comparison in luminescent properties between Zn2.4Li0.6Ga2GeO8:Mn and Zn3Li0.4Ga1.6GeO8:Mn. The comparison is helpful for readers who have interest. Thus, if possible, compare the PL properties between Zn2.4Li0.6Ga2GeO8:Mn:Mn and Zn3Li0.4Ga1.6GeO8:Mn, and give some discussion.

 Response 1: We thank the Reviewer’s suggestion. As shown in Fig. 3 and Fig. 4, the two phosphors including Zn3Li0.4Ga1.6GeO8:Mn and Zn2.4Li0.6Ga2GeO8:Mn exhibit different fluorescence properties. For example, under excitation at 337/357 nm, the Zn3Li0.4Ga1.6GeO8:Mn phosphor shows a red fluorescence output, while the Zn2.4Li0.6Ga2GeO8:Mn phosphor shows a green fluorescence output due to different lattice occupancy of Mn ions. Considering only the green afterglow characteristics, Zn2.4Li0.6Ga2GeO8:Mn phosphor exhibits brighter and long-lasting green afterglow.

Comment 2: In addition, there is no mention about the excellency of the samples prepared by the hydrothermal method compared with that prepared by the solid-state reaction. Give a short comment about this with comparing the PL intensity.

 Response 2: A good point. The Zn3Li0.4Ga1.6GeO8:Mn and Zn2.4Li0.6Ga2GeO8:Mn phosphors prepared by the hydrothermal method exhibited excitation wavelength-dependent polychromatic and multimodal fluorescence properties (Fig. 3 and 5), while samples prepared by the solid-state reaction method only exhibited red photoluminescence properties (Fig. 6).

Comment 3: In Fig. 5, the PLE spectra are not matched with the PL spectra. At 365 nm, the PLE intensity for the sample without Li+ is higher than that with Li+. But, in the PL emission, the Li-substituted sample is better than the sample without Li+. Therefore, it is necessary to check if the data is correct.

 Response 3: Thanks for the comment. Data have been confirmed and corresponding revisions have been made in the revised manuscript.

Comment 4: Is there any reason that both Li and Mn contents are fixed?

 Response 4: Thanks for the comment. We choose the best doping concentration based on literature reports[8,11].

Comment 5: In XRD patterns, the Li content is given as 20%. Actually, the Li+ concentration used is about 6.7% of Zn and about 10% of Ga. Check again the Li+ concentration.

 Response 5: Thanks for the comment. The correct form of phosphors should be Zn3Li0.4Ga1.6GeO8:Mn and Zn2.4Li0.6Ga2GeO8:Mn. They have been corrected in the revised manuscript.

Comment 6: There is no information about the doping place of Mn. Authors should give the exact information about the site where Mn ions are substituted.

 Response 6: Thanks for the comment. According to the spectral characteristics from Mn ions, we are able to infer the doping site of the Mn ions.

Comment 7: Line 111 and 115, the ‘PDF#’ expression is needed to be changed as JCPDS #.

Response 7: Thanks for the comment. Corresponding modification have been made in the revised manuscript.

Reviewer 2 Report

The manuscript named “Tuning Multicolor Emission of Zn3Ga2GeO8:Mn2+ Nanophosphors by Regulating Mn Ions Occupying Sites for Multiple Anti-counterfeiting Application” reports the hydrothermal and solid-state synthesis of the target compounds and investigation of their luminescence properties. Several comments on this manuscript are given below.

  1. First of all, I would like to commend the authors for taking their time to prepare nice-looking graphics. They are very appealing.
  2. The English should be checked since there are some grammar and punctuation mistakes (like line 212, …540 n nm…; 2E levels should be written as 2E throughout the entire manuscript, etc.).
  3. The title mentions just Mn2+ ions, whereas the manuscript deals with Mn2+ and Mn4+. This should be indicated in the title in some way.
  4. The sentence in lines 38-39 is not clear. Please rewrite.
  5. Was anhydrous Mn(NO3)2 used, or was it tetrahydrate?
  6. Line 76. I doubt the authors have used a polyethylene reactor since its melting point is 115-135 °C. Maybe polytetrafluoroethylene?
  7. Line 90. What is …certain amount…? The experimental part should be written so that no guessing is necessary.
  8. The sentence in lines 87-90 is not clear. Please explain more clearly.
  9. According to IUPAC, arbitrary units should be abbreviated as “arb. units”.
  10. Authors state that solid solutions of ZnGa2O4 and Zn2GeO4 are obtained. However, XRD patterns show that the mixtures of these compounds are obtained rather than solid solutions. Only the pattern given in Fig. 1c looks like a single-phase compound.
  11. Line 111, Zn2GeO4 crystallizes in the rhombohedral crystal structure, not in hexagonal as stated by authors. Besides, authors should give the space group of each structure for better clarity.
  12. Lines 151-153. The authors could have prepared the undoped material and measured the emission spectra to determine if the mentioned broad emission band could be assigned to matrix defects.
  13. Line 162. Since Mn4+ emission was monitored, can it be that Mn2+−O2− CTB is observed in the excitation spectrum? Can it be that it is just two severely overlapped absorption bands of Mn4+? 4A2g→ 4T1g for 327 nm and 4A2g → 4T2g for 467 nm?
  14. Lines 167-168. I am not sure if the authors can draw such a conclusion. Since emission originates from Mn2+ and Mn4+ is also possible that both these ions occupy the same lattice site.
  15. Lines 180-182. It is unclear what the authors want to say by the given sentence. Please rephrase.
  16. 3 and Fig. 4. Why were emission spectra for 980 nm excitation recorded only to ca. 700 nm?
  17. Line 205. The same comment as 13.
  18. Lines 205-208. Could authors explain why PL emission spectra are so different from PSL emission spectra? Especially in the red spectral region, where the shift of emission maxima is around 50 nm.
  19. Line 220. The same comment as 12.
  20. 5. The given emission spectra are virtually identical. Then how can it be that the emission color is so different?
  21. Lines 238-239. Why is that? One could expect an increase in the crystal field since the charge of the lattice site increases.
  22. Lines 258-260. Could the authors explain why no red PersL is observed?
  23. Lines 279-281. How could scintillation of nanomaterials be used in solid-state lighting? This sentence should be rewritten to avoid such confusion.
  24. Line 288. In line 277, the laser power is given as 0.5 W; here, it’s already 1 W. Which value is the correct one?

Author Response

Reviewer #2:

The manuscript named “Tuning Multicolor Emission of Zn3Ga2GeO8:Mn2+ Nanophosphors by Regulating Mn Ions Occupying Sites for Multiple Anti-counterfeiting Application” reports the hydrothermal and solid-state synthesis of the target compounds and investigation of their luminescence properties. Several comments on this manuscript are given below.

Comment 1:  First of all, I would like to commend the authors for taking their time to prepare nice-looking graphics. They are very appealing.

Response 1: Thank you for nice comment.  

Comment 2: The English should be checked since there are some grammar and punctuation mistakes (like line 212, …540 n nm…; 2E levels should be written as 2E throughout the entire manuscript, etc.).

Response 2: Thanks for the comment. We have carefully checked the English and corrected by our English Native speaker William Nguyen.

Comment 3: The title mentions just Mn2+ ions, whereas the manuscript deals with Mn2+ and Mn4+. This should be indicated in the title in some way.

Response 3: A very good point.. We have changed the title to “Tuning Multicolor Emission of Manganese-Activated Gallium Germanate Nanophosphors by Regulating Mn Ions Occupying Sites for Multiple Anti-counterfeiting Application” in the revised manuscript.

Comment 4: The sentence in lines 38-39 is not clear. Please rewrite.

Response 4: Thanks for the comment.  We have corrected them and made them clear.

Comment 5: Was anhydrous Mn(NO3)2 used, or was it tetrahydrate?

Response 5:  No, it is not. Aqueous solutions (0.5 M) of Mn(NO3)2 were used in the hydrothermal preparation process.

Comment 6:  Line 76. I doubt the authors have used a polyethylene reactor since its melting point is 115-135 °C. Maybe polytetrafluoroethylene?

Response 6: You are right, a polytetrafluoroethylene reactor was used.

Comment 7:  Line 90. What is …certain amount…? The experimental part should be written so that no guessing is necessary.

Response 7: Sorry for that! Typically, the ZGGO:Mn phosphors were dispersed into a mixed solution of 1.0 mL hydrochloric acid (0.2 mol/L) and 1.0 mL anhydrous C2H5OH and then the mixed solution was centrifuged and washed with deionized water and anhydrous C2H5OH for several times. Finally, the samples and PVA aqueous solution are mixed in a ratio of 1:1 to fabricate luminescent anti-counterfeiting inks. We made them clear in the revision.

Comment 8: The sentence in lines 87-90 is not clear. Please explain more clearly.

Response 8:  We corrected and made the clearer.  

Comment 9: According to IUPAC, arbitrary units should be abbreviated as “arb. units”.

Response 9:  Thanks, we corrected them.  

Comment 10: Authors state that solid solutions of ZnGa2O4 and Zn2GeO4 are obtained. However, XRD patterns show that the mixtures of these compounds are obtained rather than solid solutions. Only the pattern given in Fig. 1c looks like a single-phase compound.

Response 10: Thanks for the comment. You are right, Fig. 1a,b,d are the mixture and only Fig. 1c is the solid solution. We made them clear.

Comment 11: Line 111, Zn2GeO4 crystallizes in the rhombohedral crystal structure, not in hexagonal as stated by authors. Besides, authors should give the space group of each structure for better clarity.

Response 11: Thanks for the comment. You are right, Zn2GeO4 crystallizes are the rhombohedral crystal structure (PDF# 04-001-9915) R3Ì…. The spinel ZnGa2O4 system is of the Fd3Ì…m space group in a cubic structure. [J. Phys. Chem. C 2021, 125(39), 21780-21790].

Comment 12: Lines 151-153. The authors could have prepared the undoped material and measured the emission spectra to determine if the mentioned broad emission band could be assigned to matrix defects.

Response 12: Thanks for the comment. The emission spectra of the undoped material have been measured and shown in Fig. 4 to verify emission of matrix defects in the revised manuscript.

Comment 13:  Line 162. Since Mn4+ emission was monitored, can it be that Mn2+−O2− CTB is observed in the excitation spectrum? Can it be that it is just two severely overlapped absorption bands of Mn4+? 4A2g→ 4T1g for 357 nm and 4A2g → 4T2g for 467 nm?

Response 13: We agree. The excitation band I peaking at 357 nm originate from the Mn2+-O2− CTB and 4A24T1.

Comment 14: Lines 167-168. I am not sure if the authors can draw such a conclusion. Since emission originates from Mn2+ and Mn4+ is also possible that both these ions occupy the same lattice site.

Response 14: Thanks for the comment. If Mn2+ and Mn4+ ions both occupy the octahedral Ga3+ sites, they can only emit red light. If Mn2+ ions occupy the tetrahedral Zn2+ sites, the phosphors only exhibit green luminescence with afterglow feature. These phosphors including Zn3Li0.4Ga1.6GeO8:Mn and Zn2.4Li0.6Ga2GeO8:Mn simultaneously exhibit the red luminescence and the green fluorescence with afterglow property, so we can reasonably conclude that some Mn ions occupy octahedral sites and other Mn ions occupy tetrahedral sites in the phosphors.

Comment 15: Lines 180-182. It is unclear what the authors want to say by the given sentence. Please rephrase.

Response 15: Thanks for the comment. These results show that the red and the green PL with different features may be derived from the luminescent centers with different coordination environments.

Comment 16: Fig. 3 and Fig. 5. Why were emission spectra for 980 nm excitation recorded only to ca. 700 nm?

Response 16: Thanks for the comment. Since when the emission is too close to 980 nm it is easy to have scattering interference of 980 nm laser on the emission spectra.

Comment 17:  Line 205. The same comment as 13.

Response 17: Thanks for the comment. To demonstrate the charge transfer band, the spectra of the undoped samples were also measured for comparison as described in the revision to prove that, see Fig. 4.

Comment 18: Lines 205-208. Could authors explain why PL emission spectra are so different from PSL emission spectra? Especially in the red spectral region, where the shift of emission maxima is around 50 nm.

Response 18: Thanks for the comment. It is found that PL and PSL emission spectra are different in Fig. 3a. We know that the green luminescence (with PersL feature) and the red luminescence (without PersL characteristics) originate from the Mn2+ and Mn4+ luminescent centers, respectively. Therefore, theoretically, the red PSL could not be observed. Surprisingly, we could still observe the red luminescence, but no the green luminescence when we emptied the traps of Zn3Li0.4Ga1.6GeO8:Mn phosphor, and then is excited under 980 nm excitation. These suggests that the red fluorescence may be derived from the upconversion fluorescence of Mn ions [24]. These results show that the red and the green PL with different features should derive from the luminous centers with different coordination environments.

Comment 19: Line 220. The same comment as 12.

Response 19: Thanks for the comment. The emission spectra of the undoped material have been measured to verify emission of matrix defects in the revised manuscript.

Comment 20: The given emission spectra are virtually identical. Then how can it be that the emission color is so different?

Response 20: Thanks for the comment. The color is determined by the emission spectrum, and if observed carefully, the emission spectrum is different, especially in the 500-600 nm region in Fig. 6.

Comment 21:  Lines 238-239. Why is that? One could expect an increase in the crystal field since the charge of the lattice site increases.

Response 21: A very good question. Because Ge4+-Mn2+ or Ge4+-Zn2+ ion pair occupy the two Ga3+ sites and then lower the crystal field, the 4T2 energy level moves down into the 2E energy level of Mn4+ [24,25]. It is well known that ZnGa2O4 exhibited about 3% inversion defects, which meant that 3% of Zn2+ occupied Ga sites, and correlatively the same amount of Ga3+ occupied Zn sites.

Comment 22:  Lines 258-260. Could the authors explain why no red PersL is observed?

Response 22: Thanks for the comment. Because red PL originates from Mn4+ ions, while green PL originates from Mn2+ ions. We all know that only the tetrahedral coordinated Mn2+ ion can give a green PersL.

Comment 23:  Lines 279-281. How could scintillation of nanomaterials be used in solid-state lighting? This sentence should be rewritten to avoid such confusion.

Response 23: Right! We modified the statement.

Comment 24: Line 288. In line 277, the laser power is given as 0.5 W; here, it’s already 1 W. Which value is the correct one?

Response 24: Thanks for the comment. 1 W has been corrected in the revised manuscript.

Reviewer 3 Report

See attacched Comments (PDF file)

Author Response

Reviewer #3:

Comment 1: The authors considered that their synthesized materials are Zn3Ga2GeO8 crystals doped with Li/Mn or Mn. However, the reviewer cannot understand this. First, the authors are misunderstanding about a technical term of “solid solution” found in the Abstract at line (l.) 15, p. 1 and in the text at l. 110 and l. 120 in p. 3. Please note that the “solid solution” sometimes called “mixed crystal” or “alloy.” “Brass” is a very famous example of “solid solution” and is an alloy of “copper” and “zinc.” Its XRD pattern clearly different from their end-point metals (Cu and Zn). Therefore, it is not a physical mixture of Cu and Zn, but is another crystal clearly different from Cu or Zn. Some other examples of “solid solution” for semiconductors are In1-xGxN and Si1-xGex solid solutions. These alloy semiconductors have different “lattice parameters” obeying the well-known Vegard law and therefore, exhibit XRD peaks at different angles from their end-point materials (e.g., different angles from those of InN and GaN). Please see Fig. 1 in which we can understand that the XRD patterns of your obtained crystals seem to show they are not Zn3Ga2GeO8 crystals doped with Li/Mn (a, b, c) or with Mn only (d), but are only physical mixtures of “cubic ZnGa2O4” and “hexagonal Zn2GeO4.” [It is not a brass, but a physical mixture of Cu and Zn.]

Response 1: Thank you for your valuable comments. The XRD patterns of samples show that they are only physical mixtures of  the “cubic ZnGa2O4” and the “rhombohedral Zn2GeO4”, except (c) are a spinel-structured Zn3Ga2GeO8 solid solutions.

Comment 2: This definition promises us that some materials like “Zn2.4Li0.6Ga2GeO8:Mn (see, e.g., l. 67, p. 2)” and “Zn3Li0.2Ga1.8GeO8:Mn (see, e.g., l. 68, p. 2)” cannot be written as “ZGGO:Mn” because these phosphors contain “Li0.2” in their material formulae but not in the original definition of Zn3Ga2GeO8:0.25%Mn (ZGGO:Mn) in l. 66, p. 2. Is it OK? If so, what are “ZGGO” we can find in Fig. 1?

Response 2: Thanks for the comment. We have corrected them.

Reviewer 4 Report

The paper nanomaterials-1651001 reports an interesting study on a long lasting phosphorescence and on the optical stimulated emission in differently doped Zn3Ga2GeO8 matrix.

The general methods and analysis are adequately discussed but the rear one main point that should be addressed: the preparation of the samples is not adequate and dos not permit to reproduce similar samples. For example, it is not indicate how to achieve the  Lithium doping and the amount of it. The atmosphere of the solid state synthesis is not indicate as well. But in general, the preparation of the samples needs more attention and details.

A second important point is the high number of self- citation and/or inappropriate references: References from 36 to 52 are just indicate in a very brief sentence at the end of the paper ("In addition, the PSL, PL and scintillation of nanomaterials can be used for medical imaging [36], photodynamic activation[37-43], dosimetry [44-47], thermometry[48] and solid state lighting[49-52]. ) where most fo them are from Chen W.

If the authors wants to prepare a review on this topics, they should prepare a different article but in the present form does not give new insight to the reader

Author Response

Reviewer #4:

The paper nanomaterials-1651001 reports an interesting study on a long lasting phosphorescence and on the optical stimulated emission in differently doped Zn3Ga2GeO8 matrix.

Comment 1: The general methods and analysis are adequately discussed but the rear one main point that should be addressed: the preparation of the samples is not adequate and does not permit to reproduce similar samples. For example, it is not indicated how to achieve the Lithium doping and the amount of it. The atmosphere of the solid state synthesis is not indicate as well. But in general, the preparation of the samples needs more attention and details.

Response 1: Great questions!  The doping amount of Li is estimated value based on the material ratio of the precursors. The as-prepared powders were further calcined in a muffle furnace in air at various temperatures.

Comment 2: A second important point is the high number of self- citation and/or inappropriate references: References from 36 to 52 are just indicate in a very brief sentence at the end of the paper ("In addition, the PSL, PL and scintillation of nanomaterials can be used for medical imaging [36], photodynamic activation[37-43], dosimetry [44-47], thermometry[48] and solid state lighting[49-52]. ) where most of them are from Chen W.

Response 2: That is a good question. We cited and mentioned these applications, as our next step is to explore these applications. As suggested by the reviewer, we adjusted the refs.

Round 2

Reviewer 2 Report

Comment 5: Was anhydrous Mn(NO3)2 used, or was it tetrahydrate?

Response 5:  No, it is not. Aqueous solutions (0.5 M) of Mn(NO3)2 were used in the hydrothermal preparation process.

# The response is confusing since the Mn(NO3)2 concentration in the manuscript is 0.1 M.

Comment 13:  Line 162. Since Mn4+ emission was monitored, how can it be that Mn2+−O2− CTB is observed in the excitation spectrum? Can it be that it is just two severely overlapped absorption bands of Mn4+4A2g→ 4T1g for 357 nm and 4A2g → 4T2g for 467 nm?

#One keyword was missing in comment 13; therefore, the authors got the question wrong. My apologies.  

Reviewer 3 Report

We call GaN or AlN a “compound,” but never call it an “alloy. We can also rightly call GaAlN (Ga1-xAlxN) an “alloy (or an alloy compound).”

Similarly, we call Zn3Ga2GeO8 a “compound,” but never call it an “alloy.” This is because Zn3Ga2GeO8 is solely a compound, but not an alloy compound.

The authors did not replay to my Comment the reason why their obtained XRD trace of Fig. 1c is of a spinel-structured Zn3Ga2GeO8 compound, but not of a physical mixture of “cubic ZnGa2O4” and “hexagonal” Zn2GeO4.”

The reviewer strongly requested a comparison of their obtained powdered XRD trace of the Zn3Ga2GeO8 with the JCPDS (or PDF) card data of the ever-reported Zn3Ga2GeO8 crystal. If those peaks exactly agree in both “angles” and “relative peak heights,” the authors can then call their synthesized material to be surely “Zn3Ga2GeO8” compound, but not a physical mixture of ZnGa2O4 and Zn2GeO4 compounds.

Speaking forcibly, sample of Fig. 1d seems to be strongly Zn3Ga2GeO8 like; but, sample of Fig. 1c is to be a cubic ZnGa2O4 powder (this is because diffraction peaks at ~25, ~34 and ~48 deg are not clearly observed in Fig. 1c). Please deny the above possibility (i.e., sample of Fig. 1c is a cubic ZnGa2O4, but not a Zn3Ga2GeO8) from your observed XRD traces.

[It should be noted that the JCPDS card of Zn3Ga2GeO8 contain peaks that are at angles of almost equal to those of cubic ZnGa2O4 and hexagonal” Zn2GeO4 JCPDS cards. That is just the case as, e.g., wurtzite “Cu2ZnSnS4” compound, where its XRD trace contains diffraction peaks coming both from monoclinic “Cu2SnS3” and wurtzite “ZnS,” but those coming from a solely Cu2ZnSnS4 crystal with its own lattice parameters that are different from those of Cu2SnS3 and ZnS.]

Moreover, PL emission caused by the 4A2→4T2 transition has not been observed in Mn4+-doped oxide phosphors until now [see, e.g., ECS J. Solid State Sci. Technol. 9, 016001 (2020)]. This is in direct contrast to the case for Cr3+-doped oxide phosphors [ECS J. Solid State Sci. Technol. 10, 026001 (2021)]. Therefore, Refs. [24-27] in page 7 seem to be miscited.

Reviewer 4 Report

The authors changed the paper according to my previous criticisms. In my opinion the paper is now suitable to be published.